# Advances in Single-Cell Sequencing Technology and Its Application in Poultry Science

**DOI:** 10.3390/genes13122211

**Published:** 2022-11-25

**Authors:** Yong Liu, Shuangmin Liang, Bo Wang, Jinbo Zhao, Xiannian Zi, Shixiong Yan, Tengfei Dou, Junjing Jia, Kun Wang, Changrong Ge

**Affiliations:** 1College of Animal Science and Technology, Yunnan Agricultural University, Kunming 650201, China; 2College of Food Science and Technology, Yunnan Agricultural University, Kunming 650201, China

**Keywords:** single-cell sequencing, genome sequencing, transcriptome sequencing, epigenome sequencing, proteomic analysis, application, poultry

## Abstract

Single-cell sequencing (SCS) uses a single cell as the research material and involves three dimensions: genes, phenotypes and cell biological mechanisms. This type of research can locate target cells, analyze the dynamic changes in the target cells and the relationships between the cells, and pinpoint the molecular mechanism of cell formation. Currently, a common problem faced by animal husbandry scientists is how to apply existing science and technology to promote the production of high-quality livestock and poultry products and to breed livestock for disease resistance; this is also a bottleneck for the sustainable development of animal husbandry. In recent years, although SCS technology has been successfully applied in the fields of medicine and bioscience, its application in poultry science has been rarely reported. With the sustainable development of science and technology and the poultry industry, SCS technology has great potential in the application of poultry science (or animal husbandry). Therefore, it is necessary to review the innovation of SCS technology and its application in poultry science. This article summarizes the current main technical methods of SCS and its application in poultry, which can provide potential references for its future applications in precision breeding, disease prevention and control, immunity, and cell identification.

## 1. Introduction

Poultry eggs are the world’s largest egg supply and poultry meat is the world’s second largest meat supply. The role of poultry in human life is evident. Chicken is considered a widely available healthy meat due to its low price, fat and cholesterol content, as well as the lack of religious restrictions prohibiting its consumption. Therefore, the demand for chicken is increasing, and research into how to produce healthy and high-quality chicken has attracted more and more attention.

Cells are the basic units of life. Under a uniform genome blueprint and with spatiotemporal-specific regulations, cells are differentiated from a fertilized egg into various cells with different shapes, positions and functions, thus constituting a complete living body. Traditional sequencing technology studies the differences in tissue sources and measures the total average response of the cell population, or the genetic information of a cell that represents the significant majority of the cells, but cannot reflect the specific genetic information of a single cell. SCS uses a single cell to obtain a certain cell type’s gene sequence, transcript, protein and epigenetic expression profile information. This genetic and expression information can be associated with cell behavior through functional analyses, and the cells can be located in a certain tissue by spatial mapping. Finally, a clear picture of the heterogeneity and clonal evolution of different cell types is obtained [1,2]. SCS not only measures gene expression levels more accurately, but it also detects micro-expressed non-coding RNA. It has the advantages of special sample sequencing while also making up for its problems, such as the small amount of special sample acquisition and the inability to match traditional sequencing. The application of SCS to poultry breeding, disease prevention and control, immunity, and cell identification will be an exciting and challenging research endeavor. Hence, this paper reviewed single-cell sequencing technology and pioneering research studies in poultry science over the past few years, which will help to provide a reference for the further development of single-cell sequencing technology in poultry science.

## 2. Single-Cell Sequencing Technology

SCS consists of four steps: (1) single cell isolation; (2) nucleic acid amplification; (3) high-throughput sequencing; and (4) data analysis. Among them, the core technologies are single cell isolation and nucleic acid amplification (Figure 1).

### 2.1. Single-Cell Isolation

The SCS technique first isolates the single cells from tissue samples in order to obtain a qualified single-cell suspension. High-quality single-cell suspension is the key to successful single-cell research [3]. The basic steps of preparing the single-cell suspension are as shown below (Figure 2) [4].

Tissues are composed of cells embedded in the extracellular matrix, which are supported by the cell–cell connections of the extracellular matrix, a variety of proteins and other biomolecules. Proper digestion and the removal of these proteins and other biomolecules from cell suspensions requires specific enzymes. The digestive enzymes that play an important role in the decomposition of solid tissue are shown below (Table 1) [5].

Commercially available digestion mixes were developed and optimized to avoid problems that could arise from adding the aforementioned enzymes to tissue digestion mixes. Accutase is a protease and collagenase mix that mimics the action of trypsin and collagenase, but requires much lower concentrations when using standard enzymes. Accutase, as a mixture of enzymes with proteolytic, collagenolytic and DNase activities, has been found to improve the overall antigen preservation and have higher total cell yields in tissue digestion compared to those of other similar enzyme mixtures [15]. TrypLE is a recombinase that mimics the activity of trypsin without altering the cell surface antigen expression [16].

Enzymatic digestion is performed by introducing the digestion mixture into minced solid tissue and incubating it at a specific temperature (usually 37 °C). When selecting the digestive mixture for the preparation of the single-cell suspension, two important factors should be considered: the enzyme strength and enzyme concentration. The cell surface markers present on cells can be damaged by enzymes with high intensities or concentrations, affecting the availability of these markers as well as the viability of cells in further experiments. Therefore, to avoid these problems, mildly adherent cells, such as lymphocytes, can be isolated using mild enzymes over a short digestion period [17]. The biggest difficulty in single-cell isolation is ensuring cell viability and integrity. According to the sample state, the number of cells needed and the purpose of the analysis, a variety of single-cell capture methods have been proposed, including tissue enzymatic hydrolysis and tissue sectioning (Table 2). According to the respective advantages and disadvantages of these methods, researchers can choose an appropriate single-cell isolation method for the experimental conditions [18].

### 2.2. Nucleic Acid Amplification

#### 2.2.1. Single-Cell Whole-Genome Amplification

Whole-genome amplification is achieved by dissolving individual cells to obtain the effective amplification of genomic DNA in a single genome with high coverage. The total amount of DNA can be greatly increased without any sequence bias. Some thermocycling polymerase-chain-reaction-based methods are shown in Table 3.

DNA replicates in a semi-reserved high-fidelity manner, and base mismatches are inevitable despite the proofreading activity of DNA polymerase. In the future, the discovery and use of higher-fidelity DNA polymerase to reduce the amplification error rate will be the aim of single-cell whole-gene library construction and sequencing technology optimization.

#### 2.2.2. Single-Cell Whole-Transcriptome Amplification

After RNA extraction from isolated single cells and in single-cell isolation, the next step is to reverse transcribe the captured mRNA to cDNA, followed by conventional PCR or other in vitro transcription methods to amplify the entire transcriptome. Some methods of whole-transcriptome amplification are shown in Table 4.

All of these transcriptome amplification techniques can generate full-length mRNA information, but these techniques also have certain differences, such as in detection sensitivity, the minimum amount of starting material required, stability and application range. With the continuous development of sequencing technology, some shortcomings will be solved and optimized, such as the low coverage of WTA and the difficulty in detecting non-coding RNAs.

### 2.3. High-Throughput Sequencing

SCS mainly includes single-cell genome sequencing (scDNA-seq), transcriptome sequencing (scRNA-seq) and epigenetic sequencing, etc.

#### 2.3.1. Single-Cell Genome Sequencing

scDNA-seq is the amplification of trace whole-genome DNA in a single cell, and high-throughput sequencing is performed after obtaining a genome with high coverage to analyze the point mutation and copy number variation at the single-cell level, so as to reveal the intercellular differences and cell evolution.

#### 2.3.2. Single-Cell Transcriptome Sequencing

This method entails extracting RNA from isolated single cells, reverse transcribing the captured mRNA into cDNA, and finally amplifying the entire transcriptome by PCR or other transcription methods [57]. The final step is sequencing and the construction of the library. This sequencing technology objectively reflects gene expression and accurately distinguishes cells at different developmental stages, thereby revealing cell types, subtypes, states and developmental trajectories. By combining cell-specific DNA barcodes with specific mRNA molecular markers during reverse transcription for many sequencing technologies, the aim is to improve the sequencing accuracy and obtain high-throughput sequencing data [40]. In addition, single-cell optical phenotyping and expression sequencing (SCOPE-Seq) is a scalable technique that combines live-cell imaging with scRNA-seq to obtain images, movies or other phenotypic data from single cells by microscopy and links this information directly to genome-wide expression profiles [58].

#### 2.3.3. Single-Cell Epigenetic Sequencing

Different cells have the same DNA sequence, except if changes are made at the epigenetic level (DNA methylation, RNA methylation, histone modification, chromatin remodeling and three-dimensional conformation, etc.). Single-cell epigenetic sequencing is very important to study the temporal and spatial specificity of epigenetics. Some methods of single-cell epigenetic sequencing are shown in Table 5.

#### 2.3.4. Single-Cell Proteomic Analysis

Protein profiling on the surface of single cells enables the study of cell function at the single-cell level. With the continuous development of technology, different single-cell protein sequencing methods have emerged (Table 6).

#### 2.3.5. Single-Cell Multi-omics Combined Analysis

In recent years, SCS technology has revealed in detail the relationships between the genome, transcriptome, epigenome and proteome of a single cell, and made simultaneous multi-omics sequencing a reality. Compared with single-omics, multi-omics focuses more on the collection of complete cell information and pays more attention to time and space (Table 7).

### 2.4. Data Processing and Analysis

The rapid development of SCS technology, the growth of data volume and the increase in data complexity have posed great challenges to data analysis methods. A variety of data analysis software and processes have been developed to maximize the benefits of SCS technology and to uncover the biological information behind the data. Each step includes multiple analysis and processing methods and tools, including sequence alignment, quality control, the correction of batch effects, dimensionality reduction, cell subtype identification, proposed time series analyses and more in-depth analyses. The methods and tools commonly used for SCS analysis and processing are shown in Table 8 [76,77].

### 2.5. Common Databases for Single-Cell Sequencing Analysis

With the development of biometric analyses for big data and data mining, more and more single-cell public databases have emerged. Common databases for single-cell sequencing analyses are shown in Table 9.

### 2.6. Single-Cell Sequencing Technology Platform

#### 2.6.1. Single-Cell Isolation and Labeling Platform

Commonly used single cell isolation and labeling platforms are shown in Table 10.

#### 2.6.2. Single-Cell High-Throughput Sequencing Platform

There are many high-throughput sequencing platforms used by SCS, such as the Illumina series, BGI-seq series, Roche454, ABI Solid, Ion Proton, etc. At present, the Illumina series is the main high-throughput sequencing platform for single-cell sequencing, but other sequencing platforms are gradually proving to be capable of SCS, such as Ion Proton [112] and BGIseq-500 [113].

## 3. Application of SCS in Poultry Science

Since the advent of 10X SCS, most studies have focused on model species, such as primates and mice, but with the continuous maturity and development of the technology, it is inevitable that SCS technology will expand to non-model-species research. Analyses on non-model species from the single-cell level play an important role in promoting developmental biology as well as the origin and evolution of species. The following sections will mainly focus on the application of SCS technology in poultry research fields, such as poultry breeding, disease prevention and control, immunity, and cell identification, etc. (Table 11).

### 3.1. Application of SCS in Poultry Breeding

The most critical core technology in poultry production is breeding technology. Therefore, improving the breeding efficiency, shortening the number of breeding years and improving genetic resources are all challenges in the breeding process. With the development of biomolecular technology, poultry genetic breeding and molecular technology are increasingly integrated, and whole-genome breeding, gene-editing breeding and molecular-design breeding are prominent.

The application of SCS technology provides a new technical method for breeding practices. Estermann et al. [114] used scRNA-seq to reveal the gender differentiation in the gonads of chicken and mouse cells and the biological mechanisms underlying these differences. They identified two different support cells to the transcriptome, and with the support from the differentiation of precursor cells, derived a steroid-generated spectrum. In contrast to other vertebrates, chicken embryonic-supporting cells were not derived from the coelomic epithelium, the supporting cells of chicken embryos from the mesenchymal PAX2+/OSR1+/WNT4 +/DMRT1+ cell population. These results suggested that just as the genetic triggers of sex differ in vertebrate populations, the cellular lineages of the gonads may vary greatly. This study breaks the long-held belief that the cellular biology of gonadal development is largely conserved and provides some new insights into gonadogenesis in vertebrates (Figure 3a). Proper gonad development plays a crucial role in the reproductive development of poultry, and this study provides new clues for gonad development in poultry. Feregrino et al. [115] studied chicken limb development in unprecedented detail by single-cell transcriptome sequencing of chicken embryo limbs at three key stages of limb development. The dynamic changes in its transcriptional profile at the genome-wide level and the dynamic changes in the corresponding cells were obtained, and a series of marker genes related to the formation of different cell types was identified. The cellular and molecular mechanisms of chicken limb development were analyzed from the cellular dimension, which provided a large number of candidate genes for follow-up studies of chicken limb formation and diversity. Mantri et al. [116] combined single-cell and spatial transcriptomics with data integration algorithms to explore four key stages of Hamburger–Hamilton ventricular development in chicken hearts with a macromolecular, spatial and temporal resolution. A total of 22,315 single-cell transcriptomes and 12 spatial gene expression maps were generated, identifying 15 distinct clusters of cardiomyocyte types. The study created a hierarchical map of the developing chicken heart cell lineage and how the cells interact during development. A single-cell, spatially resolved gene expression profile was constructed, demonstrating that spatiotemporal single-cell RNA sequencing can be used to study the interplay between cell differentiation and morphogenesis (Figure 3b). Zhang et al. [117] used scRNA-seq to analyze the transcriptome of each cell group in the anterior pituitary cells of adult chickens. Four-fifths of the known endocrine cell clusters were identified and designated as prolactin, thyrotropin, adrenocorticotropin, gonadotropin and new marker genes (PRL and ensg4 in prolactin; NPBWR2 and NDRG1 in adrenocorticotropin; DIO2 in thyroid-stimulating hormone; GRP and LHB in source gonadotropin). In addition, four non-endocrine/secretory cell types were identified in chicken pituitary, including endothelial cells (expressing IGFBP7 and CFD), follicular columnar cells (expressing S100A6 and S100A10), leukocytes (expressing JCHAIN and CRIP1) and erythrocytes (expressing BCL2L1 and BF1). Among them, FS cells could express peptides (HBEGF, WNT5A, BMP4, activin, VEGFC and NPY), growth factors and progenitor/stem-cell-related genes (notch signaling component, CDH1). The results suggested that FS cell clusters act as paracrine/autocrine signaling centers and are enriched for pituitary progenitor/stem cells. The findings provided a bird’s-eye view of various aspects of the anterior pituitary gland, including heterogeneity, gene expression profiles, cell-to-cell communication and cellular composition, for a more comprehensive understanding of vertebrate pituitary biology (Figure 3c). Yamagata et al. [118] generated a chicken retinal cell map by scRNA-seq. Their analysis identified 136 cell types and 14 locations or developmental intermediates in 6 cell types in vertebrates, including photoreceptors, bipolar cells, horizontal cells, retinal ganglion cells, amacrine cells and glial cells. Using the method of integrating the reporter gene into the selectively expressed gene based on CRISPR, for Muller glial cells, it was found that the cells with different transcription were localized along the anterior posterior, dorsal ventral and central peripheral retinal axes, and the types of immature photoreceptor cells, horizontal cells and oligodendrocytes that lasted until the late embryo stage were determined. Finally, the relationship between the types of retinal cells in chickens, mice and primates was analyzed. The experimental results lay a foundation for the study of the anatomy, physiology, evolution and development of the visual system in birds, and expand the use of chicks as models of retinal structure, function and development. Sun et al. [119] detected genome-wide alternative splicing (AS) events in a variety of chicken male germ cells: embryonic stem cells (ESC), gonadal primordial germ cells (gPGC) and spermatogonial stem cells (SSC) by single-cell transcriptome sequencing screening. There were 38,494 AS events among 15,338 genes in ESCs, 48,955 AS events among 14,783 genes in gPGCs, and 49,900 AS events among 15,089 genes in SSCs. AS events are comprehensively involved in male germ cell differentiation, but show distinct patterns in ESCs, gPGCs and SSCs. AS events may regulate the expression of germ cell-specific genes NANOG, POU5F3, LIN28B, BMP4, STRA8 and LHX9, thereby affecting cell differentiation and development. Research could help open new avenues for the development of drugs for avian reproduction, stem cell biology and human consumption or to improve treatment options for patients with male infertility. Rengaraj et al. [120] combined a novel germ-cell-tracking method with time-resolved single-cell RNA sequencing (scRNA-seq) to monitor and isolate chicken germ cells at different developmental stages. The results identified sex-specific developmental stages and trajectories of chicken germ cells, with male and female trajectories characterized by the progressive acquisition of stage-specific transcription factor activity. Evolutionarily conserved and species-specific gene expression programs during germ cell development in chickens and humans were also identified, highlighting cellular and molecular features at different stages of the germ cell development. These data are valuable for studying the transcriptional landscape, cellular heterogeneity and developmental trajectories of chicken germ cells, and provide mechanistic insights into chicken germ cell development. Li et al. [121] used 5-day-old (D5) and 100-day-old (D100) Jingxing Huang chickens as their experimental materials. Based on the isolation of primary mammary muscle cells and intramuscular adipocytes, scRNA-seq was used for transcription detection. It was found that there were five myoblast subpopulations, two adipocyte subpopulations and one erythrocyte subpopulation in D5. There were one myocyte population, one adipocyte population, one erythrocyte population, one endothelial cell population and one satellite cell population in D100. The Myf5, NRXN1, RASD1 and FGFR4 genes were related to the biological process of muscle tissue development, and 19 genes, including APOA1, were related to the biological process of lipid cell IMF metabolism. These genes can be used as molecular markers to select myoblasts and IMF cells, and provide help for the breeding of chickens in terms of growth, fat deposition and other traits.

### 3.2. Application of SCS in Other Fields of Poultry

Obesity is a complex chronic disease and is currently a global epidemic. Current studies report that melatonin plays a role in preventing obesity and its complications. However, the current understanding of the melatonin-induced transcriptional dynamics of preadipocyte heterogeneity is limited. Li et al. [122] used scRNA-seq to analyze the cellular heterogeneity and transcriptional dynamics of melatonin-treated preadipocytes from both mice and chicks. The G0S2 negative cluster (G0S2-) subtype was created by systematic and sequential analyses of the transcriptomes of normal preadipocytes (Con group) and melatonin-treated preadipocytes (Mel group). G0S2 cell clusters played an important role in promoting lipolysis and inhibiting lipogenesis. By constructing G0S2-cluster-differentiation trajectories using Monocle, they identified lipid catabolic branches and networks, constructed a comprehensive single-cell-resolved differentiation roadmap for melatonin-treated preadipocytes, and provided new insights into the G0S2 cluster, which may be a major contributor to melatonin-mediated obesity (Figure 3d).

The immune system consists of cells, tissues and organs that mediate hosts’ defenses against pathogens. At present, the immunity of birds is mainly an innate immune response, while adaptive immunity has a protective effect during virus-induced avian infections. However, the important marker genes of chicken immune cells have not been clearly explored, so scRNA-seq can be used to determine chicken immune cell subsets or lineages. Dai et al. [123] found that viremia caused by avian leukemia virus subgroup J (ALV-J) resolved at 21 days post-infection (DPI); from the scRNA-seq analysis of peripheral blood lymphocytes (PBLs) from the infected and uninfected chickens (control), eight cell populations and their potential marker genes were discovered. Among them, the T cell population responded strongly and could be further divided into four subpopulations: CD4+ T cells, Th1-like cells, Th2-like cells and CD8+ T cells. ALV-J infections induce the differentiation of CD4+ T cells into Th1-like cells. The ALV-J infection significantly affected the composition of the PBL cells relative to that of the controls. The proportion of cytotoxic Th1-like cells and CD8+ T cells was increased in the peripheral blood T cell population of chickens infected with ALV-J, which may be a key factor in mitigating ALV-J infections. These results provide the basis for a systematic understanding of the function of PBL subsets and their response to viral infections under steady-state conditions. Furthermore, scRNA-seq technology has not yet been used to study the response of chicken PBL to any viral infection. SCS technology can comprehensively decode the ALV-specific cellular immune response of chickens, and then can develop vaccines or drug treatments against viruses affecting avians, which will have a huge impact on the global poultry industry. SCS is very important in dissecting immune responses. Single-cell sequencing technology can reveal immune cell specificity and function as well as key pathways for gene regulation in immune function. Additionally, it can analyze immune cell subsets and intercellular networks, explore the mechanisms of and differences in immune systems, and discover the potential functions of immune cells.

The identification of cell types can provide an in-depth understanding of cell functions. The most basic and important application of SCS is the identification of cell types. By obtaining single-cell gene expression profiles, it provides a high-precision systematic approach for the identification of cell types. Sacher et al. [124] analyzed all genes encoding extracellular matrix structural proteins (core matrix bodies) from embryonic scRNA-seq data for their ability to express different cell types. The results showed that in chicken and mouse species, the core matrix bodies had a low ability to predict undifferentiated cells during the early stages of limb development, but at later stages of development, the core matrix bodies predicted more differentiated cells than transcription factors. For distantly related species, cell-type-specific features are evolutionarily conserved, and core matrix bodies are more valuable for identifying cell types and cell states than mature cells. As embryonic tissues mature, each cell type exhibits a unique extracellular matrix or maternal type that is more suitable as a cell-type-specific transcriptional signature within a species. Therefore, using scRNA-seq technology to analyze the matrix types of different cell types and states and construct the corresponding cell maps will provide a large number of cell types and marker gene references, which are of great significance for enhancing cell function and precision medicine research.

SCS technology is a technology that has developed rapidly in recent years and is closely integrated with the field of life science. It can directly reflect the characteristics of the basic unit of life. Using this technology can provide biological heterogeneity information and also facilitate the study of low-quantity biological materials, which play a key role in biological development. With the improvement in the stability and sequencing throughput of SCS technology and the reduction in sequencing costs, this technology is now widely used in reproductive development, immunology, tumor research, disease course monitoring, mental disease research and other fields. SCS studies have also given us a deeper understanding of other organs, such as the retina, hypothalamus, cortex, gut, blood, heart, lungs and more. At present, the main research objects of SCS mainly involve humans and mammals, and there are relatively few applied research studies on poultry animals. As SCS technology deepens our understanding of various biological processes and reveals the dynamic changes in genes at the single-cell level, it will have broad implications for basic and clinical research in poultry. The application of SCS technology to poultry can enable us to better study the formation of germ cells and the early embryonic development of poultry and better guide the genetic breeding process of poultry. The heterogeneity of cells can also be analyzed for disease-infected poultry, providing a theoretical basis for disease treatment, as well as a theoretical and practical basis for adaptive vaccine development. SCS technology can also be applied to the biology of the skeletal system of poultry, by understanding the origin and formation mechanism of bones, cartilages, ligaments and other skeletal tissues. It is possible to better understand the normal physiology of the skeletal system and gain a better understanding of the pathophysiology and pathogenesis of diseases. Therefore, SCS in poultry will have many potential applications in future research.

## 4. Challenges and Prospects

Compared with traditional high-throughput sequencing, the technical difficulties of single-cell sequencing detection lie not in the sequencing itself, but in the following: first, How to capture a single active cell sample; second, In the process of library construction, how to carry out the high-quality and large-scale amplification of a very small amount of nucleic acid in a single cell to avoid nucleic acid loss and amplification offset [125]; third, How sequencing will be able to generate a massive amount of sequencing data, store data, mine, conduct analyses, conduct system integration, etc; fourth, How to improve analytical flux and save costs, how to obtain cell location information [126], and how to match the differences reflected by single-cell sequencing data with cell functions.

In recent years, with the continuous development of artificial intelligence and bioinformation algorithms, breakthroughs have been made in the field of single-cell sequencing data analysis. Xiong et al. [127] cleverly solved the problems of high dimensional, sparse and binarized single-cell ATAC-seq data by using an artificial intelligence deep learning algorithm combined with a variational autoencoder and Gaussian mixture model. Xie et al. [128] determined the differentiation trajectory of six mesenchymal cell types in normal lungs and seven mesenchymal cell types in fibrotic lungs by means of machine learning, which provided new resources for understanding the structure of fibroblasts and the role of fibroblasts in fibrotic diseases. Duan et al. [129] used model-based understanding of single cell CRISPR screening (MUSIC) to effectively analyze single-cell CRISPR screening data, and revealed the biological significance of the data. He et al. [130] proposed a deep learning imputation model with semi-supervised learning for single-cell tranomes (DISC), which used a small amount of expressed gene information and a large number of expression structures between missing expression genes to repair the gene expression distribution and to predict differential genes, the gene expression correlation and rare cell types. It provided important technical support for single-cell sequencing data analysis.

Transcriptome, apparent group or protein, and joint genome transcriptome integration analyses can reveal the effects of genetic variation on transcription [131] as well as the analytic properties of the genetic mechanism. SCS combined with chromatin accessibility analyses can build regulatory networks from the DNA to RNA to phenotype, and identify core regulatory factors strongly associated with phenotypes [132,133]. Multi-omics analyses by SCS can provide a comprehensive understanding of each cell. In the future, the direct combination of clonal structures, cell subtypes with gene variation and phenotypes, and artificial intelligence big data analysis will lay a foundation for people to explore precision medicine and precision breeding.

In recent years, the development of SCS technology has been applied to resolve gene expression distributions in cells, while generating large-scale single-cell data. The GEO database at NCBI contains a large amount of single-cell sequencing data, but this database has a fragmented storage problem compared to some dedicated databases. Therefore, there is a need to create a single-cell database for tissue-specific or disease-specific single-cell data to facilitate gene searches. However, there are currently no public platforms for single-cell data in the poultry sector, and access rights to the data are an obstacle to their effective use. In addition, the validation of one’s own data by external single-cell data is important for future single-cell data analyses. Therefore, there is a need to improve the single-cell data platform in the poultry sector by drawing on the extensive single-cell literature. By improving the single-cell database, contamination in the analyses of single-cell data can be reduced, helping researchers to more accurately uncover the biological information and mechanisms hidden in the data.

## 5. Conclusions

Single-cell sequencing (SCS) uses a single cell as the research material and involves three dimensions: genes, phenotypes and cell biological mechanisms. SCS can locate target cells, analyze the dynamic changes in the target cells and the relationships between the cells, and pinpoint the molecular mechanism of cell formation. In recent years, SCS has been successfully applied in medical and biological sciences, but its application in poultry science has rarely been reported. Therefore, SCS has great potential to be used in future studies in poultry science (or animal husbandry) learning applications. This will be one of the paths to promote the production of high-quality livestock and poultry products, disease-resistant breeding, and the sustainable development of the breeding stock industry.

## Figures and Tables

**Figure 1 genes-13-02211-f001:**
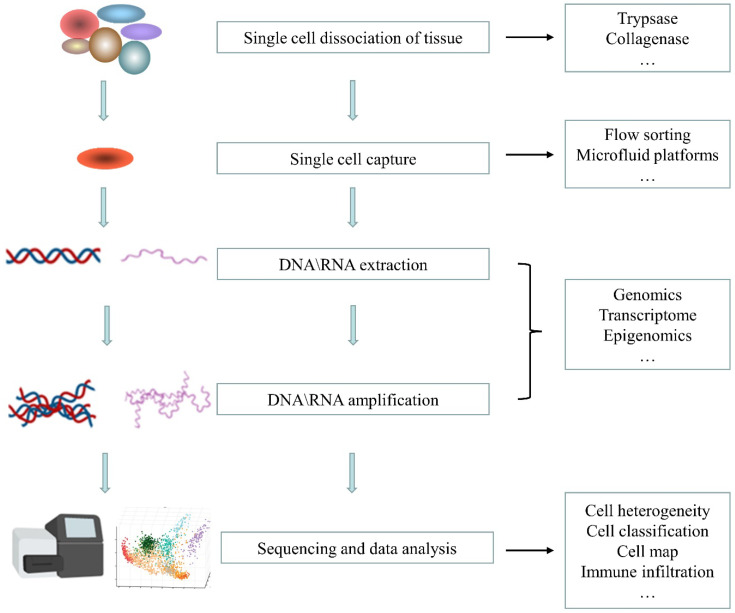
Schematic of single-cell sequencing.

**Figure 2 genes-13-02211-f002:**
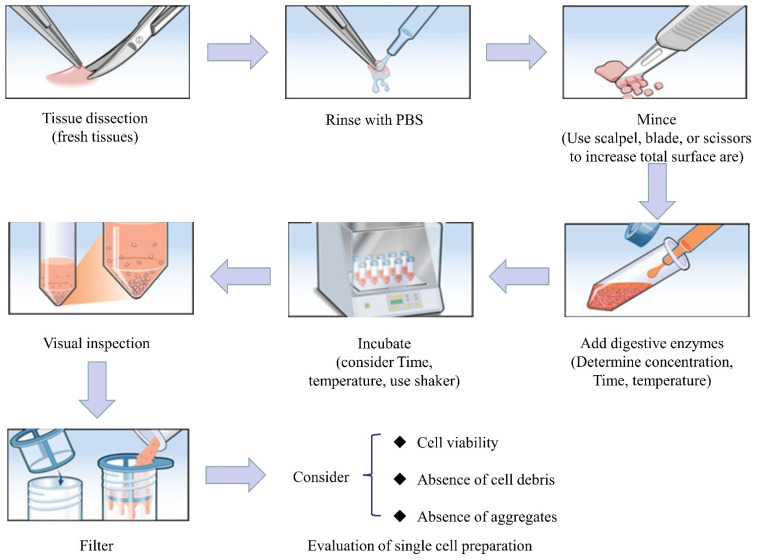
Schematic representation of the preparation of single-cell suspension from solid tissue.

**Figure 3 genes-13-02211-f003:**
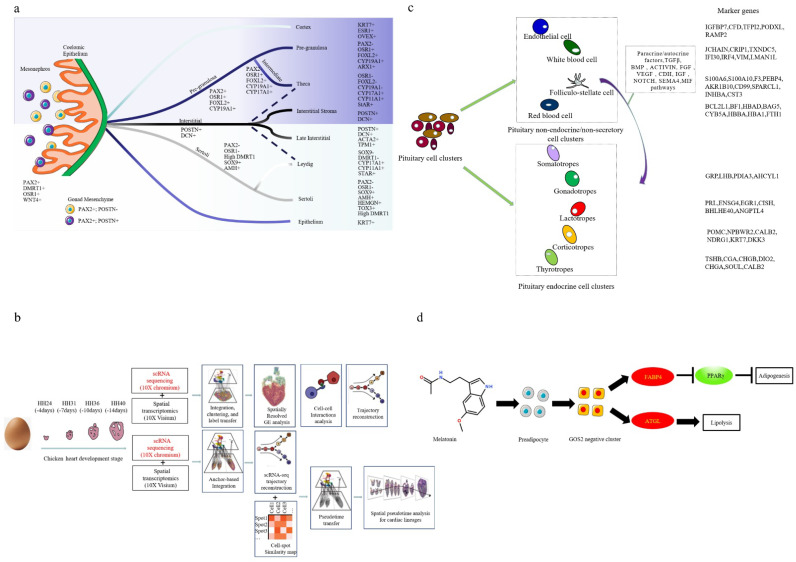
Application of SCS in poultry organ and limb development. (**a**) Schematic representation of cell lineage specification in the embryonic chicken gonad. (**b**) Analysis for scRNA-seq and spatial RNA-seq of embryonic chicken hearts at four stages of development, and schematic representation of cell lineage specification in the embryonic chicken gonad. (**c**) Chicken pituitary cell heterogeneity. (**d**) Schematic representation showing that melatonin induced preadipocyte cell fate heterogeneity and contributed to lipolysis.

**Table 1 genes-13-02211-t001:** Digestive enzymes in the breakdown of solid tissue.

Enzyme	Purpose	Attention	Ref.
Enzymes of break down the extracellular matrix			
Dispase	Cleaves attachments between cells and extra- cellular matrix.	Able to cleave specific relevant surface molecules or antigens, resulting in loss of epitopes.	[6,7]
Collagenase	Breaks peptide bonds present in collagen.	Purified collagenase is more effective.	[8]
Hyaluronidase	Cleaves glycosidic bonds in hyaluronan.	Cleaves β-1,4-glycosidic in the glycosaminoglycan.	[9]
Enzymes of break cell–cell junctions			
Trypsin	Degrades certain proteins in cell–cell junctions.	Affects cell membrane proteins. Leads to free-DNA-induced aggregation of cells.	[10,11]
Papain			[12]
Cleaves the phosphodiester linkages of the DNA main chain (Deoxyribonuclease)			
DNase-I	Degrades free DNA; prevents cell aggregation.	-CaCl_2_ acts as enzyme activator.	[13,14]

**Table 2 genes-13-02211-t002:** Single-cell isolation methods.

Methods	Description	Sorting Function	Advantages	Disadvantages	Ref.
Tissue enzymatic hydrolysis Methods					
Abundant Cells					
Serial dilution	Prepared by diluting cell populations by a series of multiples.	No	Convenient operation.	Strong dependence on the calculation of gradient dilution; easily makes errors; time-consuming; low flux rate.	[19]
Mouth pipetting	Isolate single cells with glass pipettes.	No	Convenient operation.	Identifying cells is error-prone; technically challenging; low flux rate.	[20]
FACS	Single-cell suspensions are labeled with specific fluorescent pigments. The light detector captures cell-specific signals excited by the laser to analyze cell types.	Yes	Higher accuracy, throughput and sensitivity.	Expensive equipment; damages the cells; has certain requirements for the initial number of cells.	[21,22,23]
Micromanipulation	Uses a micromanipulator to separate cells under a high-power inverted microscope.	Yes	Convenient operation; low cost.	May cause mechanical damage to targeted cells; high error rates; low flux rate.	[24]
Microfluid platforms	Separates target cells on a microscale based mainly on their characteristics.	Yes	Cell contamination rates of the microfluidic devices and reagent consumption are relatively low.	Expensive consumables; high-throughput.	[25,26,27]
Rare Cells					
Nano filter	Size discrimination on nano filters.	No	Low cost.	Cells can adhere to filters during backwash; low flux rate.	[28]
IMS	Magnetic beads are employed to bind cell surface antigens with specific monoclonal antibodies and are retained in the magnetic field.	Yes	High enrichment of rare cells.	Operation is relatively complex; low flux rate.	[29,30,31]
Cell Selector	Robotic capillary micro-manipulator.	No	Efficient isolation; high-throughput.	Expensive system.	[32]
DEP-Array	Microchip with dielectrophoretic cages.	Yes	High sensitivity.	Time-consuming; cells are deposited; low flux rate.	[33]
Tissue slice Methods					
LCM	Cells are cut from a tissue section slide with lasers.	Yes	Determines the spatial position of individual cells.	High cost, low flux rate, limited accuracy. Cell nucleus can easily be cut.	[34]
Patch clamping	Single cell is obtained from a tissue section.	Yes	Spatial context is preserved.	Contamination surrounding cells; low flux rate.	[34]

**Table 3 genes-13-02211-t003:** Single-cell whole-genome amplification methods.

Methods	Principle	Polymerase	Primer	Advantages	Disadvantages	Ref.
LA-PCR	After polymerase modifies the ends, the specific ligaments are used as the starting template, and the ligation sequences are used as primers for amplification.	Taq DNA polymerase	Random primers.	Less offset for sequence selection.	Nonspecific high amplification, low amplification efficiency, limited application, low genome coverage.	[35]
PEP-PCR	Fifteen oligonucleotides act as a primer in DNA polymerase, then the genome is randomly amplified.	Taq DNA polymerase	Random primers.	Arbitrarily primed PCR.	Nonspecific high amplification, low amplification efficiency, limited application, low genome coverage.	[36]
DOP-PCR	3′ random sequence containing 6 bp of primers randomly combined with genomic DNA to achieve the amplification of the whole genome.	Taq DNA polymerase	Degenerate oligonucleotide primers.	Suitable for CNV on chromosomes quantitatively.	Nonspecific high amplification, low amplification efficiency, limited application, low genome coverage.	[37]
MDA	Random hexamer primers and the enzyme are used to initiate replication at multiple sites simultaneously under isothermal conditions, amplified by chain substitution.	φ29 DNA polymerase	Random primers.	High amplification ability, low PCR amplification deviation, simple experiment.	Uneven coverage of the whole genome, and allelic gene loss rate up to 65%, which is not suitable for analyses of CNV.	[38,39]
MALBAC	Combines MDA with PCR.	*Bst* DNA polymerase	Random primers and common marker primers.	Simple operation, high yield, high uniformity, low amplification bias, high coverage rate.	The amplification efficiency is relatively low; the complex DNA secondary structure cannot be effectively amplified.	[40]
LIANTI	The TN5 transposon is used to randomly cut DNA for in vitro transcription and reverse transcription.	Taq DNA polymerase	Random primers.	Sequence bias reduced, high gene coverage, high accuracy of CNV detection.	False positive value of C-T base pair is higher.	[41]

**Table 4 genes-13-02211-t004:** Single-cell whole-transcriptome amplification methods.

Methods	Principle	RNA Capture	cDNA Coverage	UMI	Advantages	Disadvantages	Ref.
PloyA tail was added							
Tang RNA-seq	The polymeric T base is used as the primer to synthesize cDNA. The polymeric A base is added at the 3′ end as the binding site of the polymeric T base of the second cDNA strand.	ployA+	Full-length with 3′-biased.	NO	Can detect the full length of transcript, more sensitive and accurate.	Bias to the 3′ end high, low throughput, more expensive.	[42,43]
Quartz-seq	Inhibitory PCR is used to self-hybridize the primers to form a pan structure to reduce the by-products, and a small fragment of the second strand of cDNA is formed into a hairpin structure.	ployA+	Full-length with 3′-biased.	NO	Reduces PCR by-products, reduces contamination of small fragments.	Causes amplification bias.	[44]
SUPeR-seq	The first strand of cDNA is synthesized using random primers containing fixed anchoring sequences, which contain random nucleotide and oligonucleotide—DT. Poly adenylate and non-poly adenylate RNAs are sequenced.	ployA+ and ployA-	Full-length.	NO	RNA and genomic DNA are the least contaminated.	Low coverage rate.	[45]
MATQ-seq	The primers containing G, A and T bases are annealed for 10 cycles. After reverse transcription, the first chain of cDNAs is Ploy C, and then the G-enriched primers are used to efficiently synthesize the second chain.	ployA+ and ployA-	Full-length.	YES	High genome coverage, very sensitive.	Difficulty in detecting non-coding RNA.	[46]
5′ template-based replacement							
SMART-seq	The RNA is hybridized with a primer containing oligo(dT). Template-free C nucleotides are added to generate the first chain, and the oligonucleotide primer is hybridized with poly(C) highlight to synthesize the second chain.	ployA+	Full-length with weak 3′-biased.	NO	Good sequence coverage. Selective transcriptional isomers and SNV can be detected.	Non-chain-specific amplification, transcriptome bias, inefficient transcription of sequences.	[47]
SMART-seq2	Two to five template-free C nucleotides are added to the 3′ end of the cDNA. Then template conversion TSO is added to produce locking nucleotide modification at the 3′ end.	ployA+	Nearly full-length.	NO	No purification steps, transcript coverage is improved.	Non-chain-specific amplification, only sequencing poly(A)+ RNA, more expensive.	[48]
STRT-seq	By combining molecular markers with microfluidic techniques.	ployA+	5′ tag (TSS).	NO	High throughput, relatively cheap.	Low sensitivity, not suitable for analyses of variable splicing and allele expression.	[49,50]
SCRB-seq	The cDNA is obtained from single-cell mRNAs by using primers containing a barcode, UMI, ligand and oligonucleotide -dT, converted by template-dependent reverse transcriptase.	ployA	3′ tag (UTR).	YES	3 ‘chain information is preserved and enriched.	Low coverage rate, difficulty in detecting non-coding RNA.	[51]
Drop-seq	A droplet-based method in which each cDNA is labeled with a cell-specific barcode and UMI.	ployA+	3′ tag (UTR).	YES	Low-cost, rapid preparation library.	Requires microfluidic platforms, single-cell genes have low sensitivity.	[52]
In vitro transcription-based linear amplification							
CEL-seq	cDNA is obtained by an oligonucleotide-dT primer-containing barcode, connector and T7 promoter. cDNA polymerizations of multiple samples are homogenized for IVT.	ployA+	3′ tag (UTR).	NO	Reduced inter-sample contamination, low read length bias, chain specificity.	Severe 3 ‘bias, the high-abundance transcript is preferentially amplified.	[53]
CEL-seq2	UMI is introduced on the basis of CEL-seq, and the lengths of the barcode, connector and T7 promoter primers are shortened.	ployA+	3′ tag (UTR).	YES	Nucleic acid purification, reduced inter-sample contamination.	Severe 3 ‘bias.	[54]
MARS-seq	An automated, large-scale parallel RNA single-cell sequencing framework for sorting individual cells into 384-well plates based on FACS.	ployA+	3′ tag (UTR).	YES	Can strictly control amplification bias and marking error.	More expensive.	[55]
In Drops	It is very similar to Drop-Seq, but the hydrogel particles used In Drops also contain T7 RNA polymerase promoter.	ployA+	3′ tag (UTR).	YES	Low-cost, rapid library preparation.	Needs edmicrofluidic platform, the sensitivity of single-cell genes was low.	[56]

**Table 5 genes-13-02211-t005:** Single-Cell Epigenetic Sequencing.

Methods	Principle	Classification	Advantages	Disadvantages	Ref.
scRRBC	High-throughput bisulfite-transformed DNA methylation sequencing method is applied to the single-cell level.	High throughput.	Detection of single bases covering CpG island at the single-cell level throughout the genome.	Easy to degrade the DNA purification process, high bias, low coverage.	[59]
scATAC-seq	Sequencing nucleosome DNA in a single cell based on microfluidics or FACS.	High throughput.	Achieves genome-wide open chromatin sequencing at the single-cell level.	Cell capture, lysis, transposition and PCR using microfluidic chips.	[60]
scCOOL-seq	The integration, optimization and enhancement of genome-wide nucleosome mapping, DNA methylation sequencing and genome-wide bisulfite sequencing.	High throughput.	Better covers whole genome, solves the problem of insufficient effective data due to the enrichment of mitochondrial fragments.		[61]
CoBATCH	Protein A-TN5 is used to identify and cleat antibody-bound genomic regions and is used in conjunction with bar-coded single-cell technology.	High throughput.	Significantly improves the efficiency of chips, achieves high-throughput labeling of single cells.		[62]

**Table 6 genes-13-02211-t006:** Single-Cell Proteomic Analysis.

Methods	Principle	Advantages	Ref.
PLAYR	Proteins and RNA labeled with different metal isotopes by antibodies and probes are measured by flow cytometry with mass spectrometry to analyze the proteomes and transcripts.	Low cost, detection cell number is large, more than 40 kinds of mRNA and protein can be detected.	[63]
CITE-seq	Oligonucleotide-labeled antibodies and short oligonucleotide-labeled magnetic beads are used to bind cell surface protein and cytoplasmic mRNA, respectively. RNA and antibody labels are amplified and separated by size for quantitative analyses of proteins and transcripts.	About 100 proteins and tens of thousands of RNA transcripts can be detected, but compared with PLAYR, the number of cells detected at a time is smaller.	[64]
REAP-seq	Similar to cite-Seq technology, oligonucleotide cross-linked antibodies are used to detect cellular protein and transcript levels based on sequencing technology.		[65]

**Table 7 genes-13-02211-t007:** Single-Cell Multi-omics Combined Analysis.

Methods	Classification	Principle	Advantages	Ref.
DR-seq	Transcriptome and genome	Parallel sequencing of the genome and transcriptome, which amplifies DNA and RNA from lysate after single-cell lysate.	Reduces nucleic acid loss and cross-contamination.	[66]
G&T-seq		Biotinylated oligo-dT primers are used to capture RNA released by the lysis of a single cell, which is separated from DNA by streptavidin-coated magnetic beads. The RNA captured on the magnetic beads is amplified by Smart-seq2. The DNA is amplified by multiple displacement amplification.	Avoids cross-contamination.	[67]
TARGET-seq		The single-cell lysate and the protease are inactivated at high temperature, then specific primers of cDNA and gDNA are added to the mixture, followed by reverse transcription and PCR.	Can achieve parallel, non-bias and high-sensitivity whole-transcriptome mutation analysis, which reduces the complexity of library construction.	[68,69]
scM&T-seq	Transcriptome and epigenome	Based on G&T-seq technology, genomic DNA is treated with bisulfite to convert unmethylated cytosine to uracil, and then amplified and sequenced to determine methylated groups.	The relationship between single-cell DNA methylation heterogeneity and specific gene expression differences is provided.	[70]
SNARE-seq		The accessible genomic loci captured in the permeable nucleus by the Tn5 transposable enzyme were packaged in the same droplet as the mRNA from the single nucleus. Design “splint oligonucleotide” complements the 5 ‘-end transposition insertion sequence and ends with Ploy A, which is captured by the Ploy T bead. The captured mRNA and fragmented gDNA are released by heating the droplets.	Accessible nuclear chromatin and mRNA expression sequencing, enables more accurate identification of various cell types.	[71]
scCAT-seq		After single-cell lysis, the nucleus and cytoplasm are separated. Reverse transcription of cytoplasmic components is performed based on Smart-seq2. The Tn5 transferase and vector-DNA-mediated experimental protocol are used to amplify endogenous DNA and vector DNA in two steps.	Chromatin accessibility and transcriptome within a single cell can be detected simultaneously to study their regulatory relationships and identify transcription factors.	[72]
PLAYR	Transcriptome and proteome	A technique for high multiple quantification of transcripts in a single cell by flow cytometry and mass spectrometry, allowing simultaneous staining of standard protein antibodies.	More than 40 different mRNA and proteins can be quantitatively analyzed simultaneously.	[73]
scTrio-seq	Genome, DNA methylome, and transcriptome	The nucleus is separated by centrifugation, and the membrane is selectively cleaved to separate the mRNA in the cytoplasm from the genomic DNA in the intact nucleus. Genomic DNA is tested for methylation groups using modified sulfite treatment and sequencing methods.	Genomic copy number variation (CNV), DNA methylation and transcriptome of a single cell can be analyzed simultaneously.	[74]
scNMT-seq	Nucleosome, methylation and transcription	Based on scM&T-seq, physical separation of DNA and RNA is performed prior to the bisulfite conversion step. The transcriptome of the cells is analyzed using Smart-seq2 sequencing. Chromatin accessibility and DNA methylation are detected by nucleosome occupancy and methylation sequencing.	Combined analysis of transcriptome, methylation and chromatin accessibility.	[75]

**Table 8 genes-13-02211-t008:** The methods and tools commonly used for SCS analysis and processing.

Name	Classification	Core Algorithm	Main Goal/Function	Advantages	Ref.
TopHat	Comparison of data	Bowtie	Short RNA sequences above 75 bp in length are compared with the reference genome to find a match, and selective splicing of exons is performed.	Small memory, high accuracy, low error tolerance.	[78]
STAR		Maximal mappable prefix	Direct selection of discontinuous sequences for comparison.	Fast running speed.	[79]
SCnorm	Data Standardization		Normalization of sequencing data using quantile regression.	Improved principal component analyses and identification of differentially expressed genes.	[80]
Scran		PCA,SCE algorithm	Data sets are easily standardized and shared.	Suitable for a variety of sequencing methods, has a comprehensive function.	[81]
	Interpolation	AutoImpute	Finds missing values by learning the intrinsic distribution and patterns of scRNA-Seq data.	Interpolation of the largest data set without memory consumption.	[82]
	Correction of batch effects	Scanorama	Batch effect corrections using matched information.	Does not depend on the order of the data set, reducing the search time.	[83]
Scater		PCA,t-SNE	Focuses on data quality control and data visualization, applicable for toscRNA-seq.	Provides a rich suite of plotting tools for single-cell data and a flexible data structure that is compatible with existing tools, allowing data sets to be easily standardized and shared and batch effects to be identified and removed.	[84]
Scanpy	Dimensionality reduction	PCA,t-SNE,UMAP	Large data sets are available. Facilitates data exchange between different laboratories.	Comprehensive functions, supports UMAP dimension reduction.	[85]
Seurat	Cell subtype identification	PCA,t-SNE	Achieves unbiased identification of shared gene–gene correlations across data sets, as well as the alignment of canonical correlation vectors using nonlinear ‘warping’ algorithms.	Suitable for a variety of sequencing methods, comprehensive functions.	[86,87]
M3Drop		Michaelis-Menten model	Takes advantage of the prevalence of zeros (dropouts) in scRNA-Seq data to identify features. Sets of genes are often reduced through feature selection, only removing genes subject to technical noise.	Good data compatibility, suitable for all kinds of transcriptome sequencing data.	[88]
Wishbone	Proposed time series analysis	PCA,t-SNE	Uses the top diffusion components to construct graphs, capturing the major geometric structures in the data, while removing small fluctuations likely resulting from measurement noise.	Bifurcating branches are used to identify single- cell trajectories.	[89]
Monocle		DBSCAN, Louvain algorithm	Expression levels into relative transcript counts, applicable toscRNA-seq.	The relative quantification can be done accurately without control experiments.	[90]

**Table 9 genes-13-02211-t009:** Common databases for single-cell sequencing analysis.

Name	Organism	Classification	Contents	Website	Ref.
HTCA	Human	Adult and fetal phenotype mapping multi-omics single-cell database.	Interactive database of 3000 scRNA-seq samples containing in-depth phenotypic profiles of 19 healthy adult and fetal tissues.	http://www.htcatlas.org (accessed on 22 November 2022).	[91]
TISCH	Human	Tumor Microenvironment Single-Cell Database.	A total of 2045746 cells from 79 data sets and 28 cancer types.	http://tisch.comp-genomics.org (accessed on 22 November 2022).	[92]
TEDD	Human, mouse, etc.	Single-cell transcriptome and chromatin accessibility database for tissue and organ development.	RNA sequencing data from 2760 samples from humans and multiple model animals, and 5.1 million single-cell sequencing data.	https://TEDD.obg.cuhk.edu.hk/ (accessed on 22 November 2022).	[93]
ABC portal	Human, mouse	Shared database of blood cells and immune cells.	A total of 198 sets of human and mouse blood, immune-related single-cell transcriptome data sets.	http://abc.sklehabc.com (accessed on 22 November 2022).	[94]
Cancer SCEM	Human	Tumor Single-Cell Database.	A total of 208 single-cell RNA-Seq data sets of 20 human tumors.	https://ngdc.cncb.ac.cn/cancerscem/index (accessed on 22 November 2022).	[95]
HCA	Human	Cell Mapping Database.	Single-cell data from laboratories around the world.	https://data.humancellatlas.org/ (accessed on 22 November 2022).	[96]
Jingle Bells	Human	Immune and non-immune cell database.	Classification of single-cell data into immune and non-immune categories based on the single-cell literature.	http://jinglebells.bgu.ac.il/ (accessed on 22 November 2022).	[97]
CancerSEA	Human	Tumor single-cell functional state mapping database.	Cancer Single-Cell State Atlas.	http://biocc.hrbmu.edu.cn/CancerSEA/ (accessed on 22 November 2022).	[98]
scTPA	Human, mouse	Single-cell gene expression database for pathway activation signature.	Single-Cell Transcriptome Biopathway Annotation Tool.	https://www.sctpa.ca/ (accessed on 22 November 2022).	[99]
PanglaoDB	Human, mouse	Annotated database of cell fractions.	1368 sets of human and mouse single-cell transcriptome databases.	https://panglaodb.se/index.html (accessed on 22 November 2022).	[100]
CellMarkrer	Human, mouse	Database of cell marker information, tissue types, cell types, etc.	13605 Marker genes in 158 human tissues and 467 cell types.	http://bio-bigdata.hrbmu.edu.cn/CellMarker/ (accessed on 22 November 2022).	
BloodSpot	Human, mouse	Blood Cell Database.	Single-cell transcriptome data for health and blood disorders.	http://servers.binf.ku.dk/bloodspot/ (accessed on 22 November 2022).	
scRNASeqDB	Human	Human Single-Cell Gene Expression Database.	Covers 200 cell lines and 14,000 samples.	https://bioinfo.uth.edu/scrnaseqdb/ (accessed on 22 November 2022).	[101]
Single Cell Portal	Human, chicken	Understanding how cells and subsets of cells aggregate the database.	Collection of 3.4 million cell species	https://singlecell.broadinstitute.org/single-cell (accessed on 18 November 2022).	[102]

**Table 10 genes-13-02211-t010:** Single-cell isolation and labeling platform.

Platforms	Introduction	Advantages	Disadvantages	Applications	Ref.
C1^TM^ Single-Cell are automatically prepared system	Microfluidic technology is used to complete the whole process of cell capture, lysis, reverse transcription and pre-amplification on the same chip.	96 single sperm cells can be captured simultaneously.	High cost, complex operation.	Applications in reproductive development, stem cell differentiation, validation of biomarkers and silenced gene expression with RNA interference.	[97]
ICELL8 Single-Cell System	The cells to be measured ware captured, and an 8-channel nozzle is used to spray nanoscale onto the porous nanoscale chip, so that all kinds of cells stay in a single hole on the chip.	The process is simple, more cells can be separated each time (500~1000).	Cell capture efficiency is only 30%.	Suitable single-cell full-length transcriptome study.	[103]
BD Rhapsody^TM^ Single-Cell Analysis System	Based on microfluidic chip technology, single cells are captured on magnetic microspheres and specific molecular labels are attached to each transcript of single cells.	Libraries can prepare 100~10,000 single cells at a time; the detection range can be focused on target genes.	Libraries cannot be built for rare or unlabeled cells.	It can be applied to the analysis of cell and cell subsets’ expression characteristic clustering and marker screening.	[104]
Chromium ^TM^ System	Constructs a reagent delivery system to separate cells or nuclei; sequencing libraries are prepared in parallel so that all fragments produced by each droplet are labeled with a common molecular label.	A total of 80000 cells can be captured at one time; the capture rate of single sample cell is up to 65%.	Cell activity requirement is greater than 90%.	Suitable for genome assembly to obtain large fragments of genetic information.	[105,106,107]
split-pool ligation-based transcriptome sequencing (SPLiT-seq)	Cells do not need to be pre-isolated, and the cells themselves are seen as a natural isolation chamber for RNA. In each round of split pooling, fixed cells or nuclei are randomly distributed into wells to label individual transcriptome-specific barcodes.	Transcriptional analysis of hundreds of thousands of fixed cells or nuclei in a single experiment.	Sequencing costs can be high.	Identifies gene expression changes associated with the pathogenesis of complex diseases, such as Parkinson’s disease or cancer.	[108,109]
10X Geneomics	The single-cell suspension and labeled gel beads are encapsulated in droplets through a microfluidic chip for reverse transcription reaction, and the single-cell cDNA library is constructed and sequenced, then through data analysis, the mRNA sequence is identified	High cell-capture efficiency, fast cycle time, low cost and easy to operate.	Only the 3′ end is sequenced, and the coverage is relatively low.	It is used in research fields, such as Human Cell Atlas construction, development, immunity and disease.	[110,111]

**Table 11 genes-13-02211-t011:** Application of SCS in poultry science in the past three years.

Research Content	Method	Periodical and Year	IF	Ref.
Insights into Gonadal Sex Differentiation Provided in the Chicken Embryo	scRNA-seq	Cell Reports, 2020	8.109	[114]
Developing chicken limb	scRNA-seq	BMC Genomics, 2019	3.594	[115]
Early to late four-chambered heart stage development.	Spatiotemporal sigle-cell RNA sequencing	Nature communications, 2021	12.12	[116]
Bird’s-Eye View on Vertebrate Pituitary	scRNA-seq	Front Physiol, 2021,	4.566	[117]
Cell atlas of the chicken retina	scRNA-seq	Elife, 2021	8.146	[118]
Alternative Splicing Events during Chicken Male Germ-Line Stem Cell Differentiation.	scRNA-seq	Animals, 2021	2.852	[119]
Identification of diverse cell populations in skeletal muscles and biomarkers for intramuscular fat	scRNA-seq	BMC Genomics, 2020	3.594	[120]
Preadipocytes reveals induced by melatonin the cell fate heterogeneity.	scRNA-seq	Journal of Pineal Research, 2021	14.528	[121]
Chicken peripheral blood lymphocyte response to ALV-J infection assessed	scRNA-seq	Front Microbol, 2022	--	[122]
Changes in expression during hair-cell development	Single-cell proteomics	Elife, 2019	8.146	[123]
Single-cell sequencing of the bursa of Fabricius highlights the IBDV infection mechanism in chickens	scRNA-seq	Cell Bioscience, 2021	9.072	[124]
Dying hair cells in the avian cochlea	scRNA-seq	Cell Reports, 2021	8.109	[125]

## Data Availability

Not applicable.

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
