# Peer review of "Advances in Single-Cell Sequencing Technology and Its Application in Poultry Science"

_genes, 2022, doi:10.3390/genes13122211_

Round 1

Reviewer 1 Report

The manuscript made a comprehensive review of Single-cell sequencing (SCS), from the step by step technology to the pioneering application in poultry science. Many literatures were cited for presenting the latest development or innovation in method, platform, analysis and application of SCS, which can provide potential value in follow-up research on precision breeding, disease prevention and control in poultry science. Therefore the review is worthy to be considered for acceptance for publication in the journal.

Several minor points could be polished as follows:

(1)   line 169: humans could be replaced by primates;

(2)   line 180: problems could be substituted by challenges;

(3)   Line 178-270: the section could be separated into several paragraphs.

(4)   Whether all the schematic representations were well cited.

(5)   Advantage of LA-PCR and PEP PCR could be shortened to phrase or words in Table 3.

Reviewer 2 Report

Thanks for your effort to this field. Here are some comments.

1. This review has listed most common SCS analysis software in the table 8. But to some extent, still lacking details about these various software. e.g. what's the difference between each software, the main goal/function of these software.

2. Lacking details about what's the challenges in SCS data analysis, like contamination and doublets issues. There are also many software developed to address these issues.

3. Are there any database support SCS data analysis in Poultry field, since we have so many related database for human studies? Is this also another challenge in this field?

4. As a review to a wider audiences, food safety issue with the development of poultry genetic breeding and molecular technology should be also concerned.
